# Usable Information and Evolution of Optimal Representations During Training

**Michael Kleinman, Daksh Idnani, Alessandro Achille & Jonathan C. Kao**
University of California, Los Angeles
{michael.kleinman, dakshidnani}@ucla.edu, achille@cs.ucla.edu,
kao@seas.ucla.edu

## Abstract

We introduce a notion of usable information contained in the representation learned by a deep network, and use it to study how optimal representations for the task emerge during training, and how they adapt to different tasks. We use this to characterize the transient dynamics of deep neural networks on perceptual decision-making tasks inspired by neuroscience literature. In particular, we show that both the random initialization and the implicit regularization from Stochastic Gradient Descent play an important role in learning minimal sufficient representations for the task. If the network is not randomly initialized, we show that the training may not recover an optimal representation, increasing the chance of overfitting.

## 1 Introduction

An important open question for the theory of deep learning is why highly overparametrized neural networks learn solutions that generalize well even though models can in principle memorize the entire training set. Some have speculated that neural networks learn minimal but sufficient representations of the input through implicit regularization of Stochastic Gradient Descent (SGD) (Shwartz-Ziv & Tishby, 2017; Achille & Soatto, 2018), and that the minimality of the representations relates to generalizability. Follow-up work has disputed some of these claims (Saxe et al., 2018), leading to an ongoing debate on the optimality of representations and how they are learned during training. Here, we design a simple task to empirically study how representations are formed during training, and how implicit regularization from SGD and initializations affect the resulting representations in deep networks.

Investigations into the optimality of representations have typically used information-theoretic reasoning. Most previous information-theoretic studies of deep learning have simultaneously studied the amount of information a representation contains about the input and output using mutual information. However, when the mapping from input to representation is deterministic, the mutual information between the representation and input is degenerate (Saxe et al., 2018; Goldfeld et al., 2018). Rather than study the mutual information in a neural network, here we instead define and study the "usable information" in the network, which measures the amount of information that can be extracted from the representation by a learned decoder, and is scalable to high dimensional realistic tasks. We use this notion to quantify how relevant and irrelevant information is represented across layers of the network throughout the training process. Here, we propose simple tasks that allow us to characterize the usable information that representations contain, enabling us to address these questions.

Our simple task was inspired by decision-making tasks in neuroscience, where inputs and outputs are carefully designed to probe specific information processing phenomena. In particular, we primarily focus on a particular task we refer to as the checkerboard (CB) task (Chandrasekaran et al., 2017; Kleinman et al., 2019). In the CB task, one discerns the dominant color of a checkerboard filled with red and green squares. The subject then makes a reach to a left or right target whose color matches

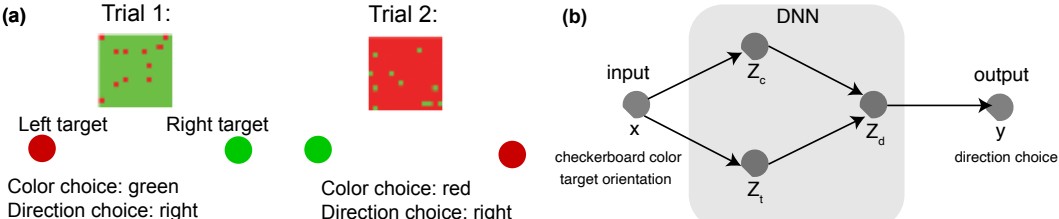

Figure 1: **(a)** Checkerboard task. Given two binary target locations (left or right) with randomly selected binary colors (red or green), one has to discern the dominant color in the checkerboard and reach to the target of the dominant color. On every trial, there is a correct color and direction choice. However, the identities of the left and right targets are random every trial, decoupling the direction and color decision. **(b)** We trained a deep neural network to perform the task by specifying the proportion of green and red squares on the checkerboard, as well as two scalars denoting the colors of the left and right target. The network was trained to output the correct direction choice. As only the direction, but not the color choice, was reported, given a representation of the correct direction choice $Z_d$, the network does not need to represent the color choice $Z_c$ in deeper layers. $Z_t$ is the representation of the target orientation.

the dominant color in the checkerboard (Fig 1a). This task therefore involves making two binary choices: a color decision (i.e., reach to the red or green target) and a direction decision (i.e., reach to left or right). Critically, the colors of the targets (red left, green right; or green left, red right) is random on every trial. The direction decision output is conditionally independent of the color decision, as detailed further in Fig 1b and section C.4, even though the color information needs to be used to solve the task. This task allows us to evaluate how both of these components of information are represented through training and across layers.

We used this task and extensions to study the evolution of minimal representations during training. If a representation is sufficient and minimal, we refer to this representation as optimal (Achille & Soatto, 2018). Our contributions are the following. **(1)** We introduce a notion of usable information for studying representations and training dynamics in deep networks (Section 2). **(2)** We used this notion to characterize the transient training dynamics in deep networks by studying the amount of usable relevant and irrelevant information in deep network layers. We found that training with SGD led to network solutions with minimal representations in later layers (Section 3.1. This adds to literature suggesting that SGD results in minimal representations of input information (Achille & Soatto, 2018; Shwartz-Ziv & Tishby, 2017). **(3)** We examined how pretraining on a related but different task affected asymptotic network representations (Section 3.2). When the network was initialized to contain usable information in later layers about a quantity it did not need to represent, SGD did not result in a minimal representation. Rather, SGD leveraged the existing representations to solve the new task, leading to representations that were similar to the initialization.

## 2   Usable information in a representation

A deep neural network consists of a set of $\ell$ layers, with each layer forming a successive representation of the input. A representation $Z_\ell$ may store information in a variety of ways. It may be that a complex transformation is required to readout the information, or it may be that a simple linear decoder could readout the information. In both cases, from an information-theoretic perspective, the same information is contained in the representation, however, there is an important distinction regarding how "usable" this information is. Information is usable if later layers, which comprise affine transformations and element-wise nonlinearities, can use the representation to solve the task. Equivalently, usable information should be decodable by a separate neural network also employing affine transformations and element-wise nonlinearities.

Formally, we define the usable information that a representation $Z$ contains about a quantity $Y$, which may refer to the output or a component of the input, as:

$$I_u(Z; Y) = H(Y) - L_{CE}(p(y|z), q(y|z)). \tag{1}$$

Here, $H(Y)$ is the entropy, or uncertainty, of $Y$, and $L_{CE}$ is the cross-entropy loss on the test set of a discriminator network $q(y|z)$ trained to approximate the true distribution $p(y|z)$. Our definition is motivated in the following manner. The test set cross-entropy loss approximates how much uncertainty there is in the output $Y$ given $Z$ and the discriminator. A low loss implies that there is low uncertainty in $Y$ given $Z$, or that the discriminator can extract a lot of "information" about $Y$ from $Z$. If the logarithm in the cross-entropy loss is base 2, it has the units of bits. If nearly all of the output classes $Y$ were the same, there would be little uncertainty in $Y$ to begin with, so it is important to know the amount of uncertainty in $Y$ given $Z$ with respect to the initial uncertainty in $Y$. What is most relevant is the amount of remaining uncertainty in $Y$ given $Z$. Thus we use the difference in uncertainty $H(Y) - L_{CE}$ as the amount of "usable information" that $Z$ contains about $Y$, as shown in our definition in Equation 1.

This definition is appealing to study representations, in part, because it can be computed from samples of $Z$ and $Y$, and is a quantity that is comparable through network training. We estimate $L_{CE}$ using a small neural network that learns a distribution $q(y|z)$. To train the network, we sample activations $Z$ and the quantity $Y$ and learn $q(y|z)$ by minimizing the cross entropy loss on a training set. We then evaluate the $L_{CE}$ on the test set (Equation 1). Details of training the neural network we used for decoding are in Appendix C.1. We also show in the Appendix that the usable information is a lower bound on the mutual information (Appendix B.1). Importantly, usable information also is not constrained by the data processing inequality; that is, information can be made more "usable" by transformations to later layers, consistent with the representation learning view that later layers are forming improved representations of the inputs (Xu et al., 2020).

## 3 Experiments

Our goal was to characterize how optimal representations are formed through SGD training and impacted by an initialization. We trained multiple network architectures on tasks and assessed the usable information in representations across layers and training epochs. Within an architecture and task, all hyperparameters were kept constant throughout experiments.

We trained three different network architectures, 'Small FC': 5 layers, with $10 - 7 - 5 - 4 - 3$ units in each layer, 'Medium FC': $100 - 20 - 20 - 20$, and 'Large FC': $1000 - 20 - 20 - 20$. Small FC and Large FC were networks used in recent literature (Shwartz-Ziv & Tishby, 2017; Saxe et al., 2018). Our networks were fully-connected and used the relu activation. We trained the networks using SGD with a constant learning rate to perform the CB task, described in detail in Appendix C.2. The hyperparameters used for all the experiments are listed in Appendix C.3.

In our CB task experiments, we quantified the usable color and direction information in the hidden representation, $Z_\ell$. In the $n = 2$ CB task, the color information represents half of the input information. We emphasize that, unless otherwise specified, the network was only trained to output the correct direction choice, so given a representation of the direction, a representation of the color choice is irrelevant. Therefore, a minimal representation should not include information about the color choice, since it is not necessary to represent given a representation of the direction decision. To make the task more complex, we also generalized the CB task to have $n = 10$ and $n = 20$ targets.

### 3.1 SGD with random initialization results in minimal sufficient representations in the CB task

We first assessed the optimality of network representations with random weight initializations by training Small FC networks on the CB task using $n = 2$ colors (Fig 2a). In random initializations, the initial weights do not contain information about the dataset. We computed the usable color and direction information across layers of the neural network and epochs of training. In our plots, later layers are denoted by darker shades. In deeper layers, there was a decrease in usable color information, corresponding to more minimal representations. After training, the asymptotic representation in the last layer contained zero usable color information and 1 bit of usable direction information. To visualize this minimal sufficient representation, we plotted the activations of the 3 units in the last layer of the Small FC network for different inputs. These visualizations are labeled by the correct color (red and green) and direction (cross or circle). In the asymptotic representation, representation of the input color is overlapping (red and green), while the representation of the direction output is separable (crosses and circles), forming a minimal sufficient representation.

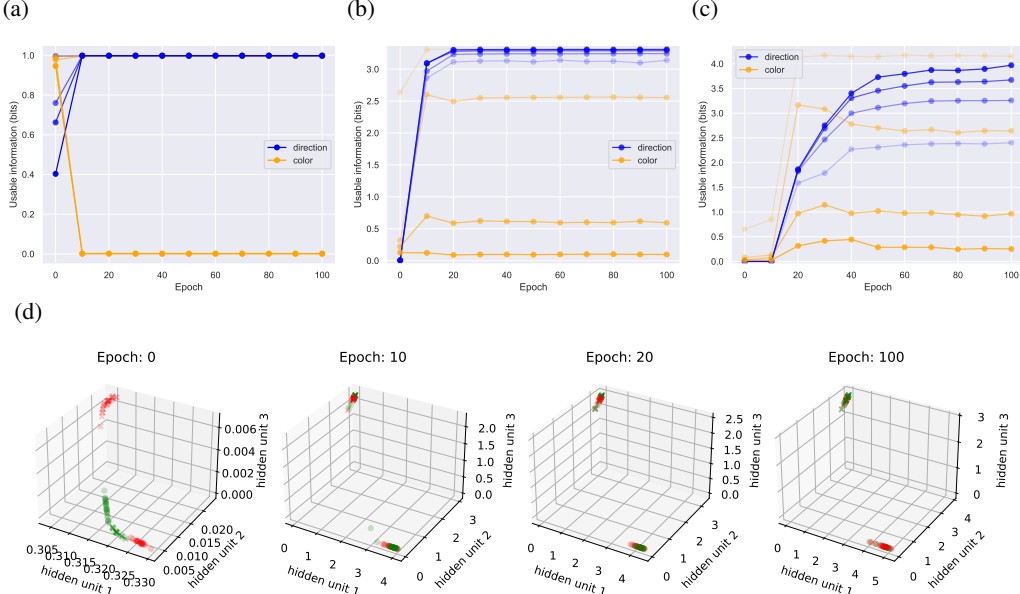

Figure 2: **SGD with random initialization leads to minimal representations. (a)** Small FC network trained on the $n = 2$ checkerboard task. Max usable direction and color information: 1 bit. This network was trained without regularization for 100 epochs using SGD with a learning rate of 0.05 and batch size of 32. Blue (orange) lines correspond to usable information about the direction (color) decision in the representation. Darker shades of color correspond to deeper layers in the network. In the asymptotic representations, we observed that direction information was high across layers, while color information decreased in the later layers.The usable color information was approximately zero in the last layer of the Small FC network. **(b)** Medium FC network trained with $n = 10$ checkerboard colors. Max usable direction and color information: 3.32 bits. In the last layer, there is nearly zero usable color information. Across layers, there is a decrease in usable color information, and an increase in usable direction information. **(c)** Medium FC network trained with $n = 20$ checkerboard colors, a batch size of 128 and a learning rate of 0.5. Max usable direction and color information: 4.32 bits. In the later layers (darker shades) there is small usable color information, but large usable direction information. **(d)** Visualization of the activations of the last layer of Small FC from (a) at epochs [0, 10, 20, 100], where the correct color choice is denoted by the marker color (red or green) and the correct direction choice is denoted by marker shape (crosses or dots). After training the crosses and dots are overlapping, corresponding to nearly zero usable color information and nearly 1 bit of direction information. This is a minimal and sufficient representation to solve the task.

To test if this observed minimality was a result of our simple task, we extended the CB task to a variant with *n* input checkerboard colors, with *n* corresponding output direction classes. We trained networks using a larger architecture (Medium FC). We show results for $n = 10$ and $n = 20$ classes in Fig 2b,c. We observed similar phenomena to the $n = 2$ case: there was decreasing usable color information in deeper layers, and nearly zero color information in the last layer's representation. In contrast, there was significant usable direction information across all layers in the asymptotic representation, with usable direction increasing for deeper layers. We validated our results using different random initializations (Figures 6, 7, 8).

These results show that, for a simple task with SGD and random initialization, minimal sufficient representations emerge through training. Asymptotic representations were sufficient to perform the task, but contained less usable color information in deeper layers, approaching zero color information in the last layer. We observed it was possible for the network to solve the task with nearly zero usable color information in its last layer across all training (Fig 2b,c). Additional runs are shown in Figures 9, 10, and 11.

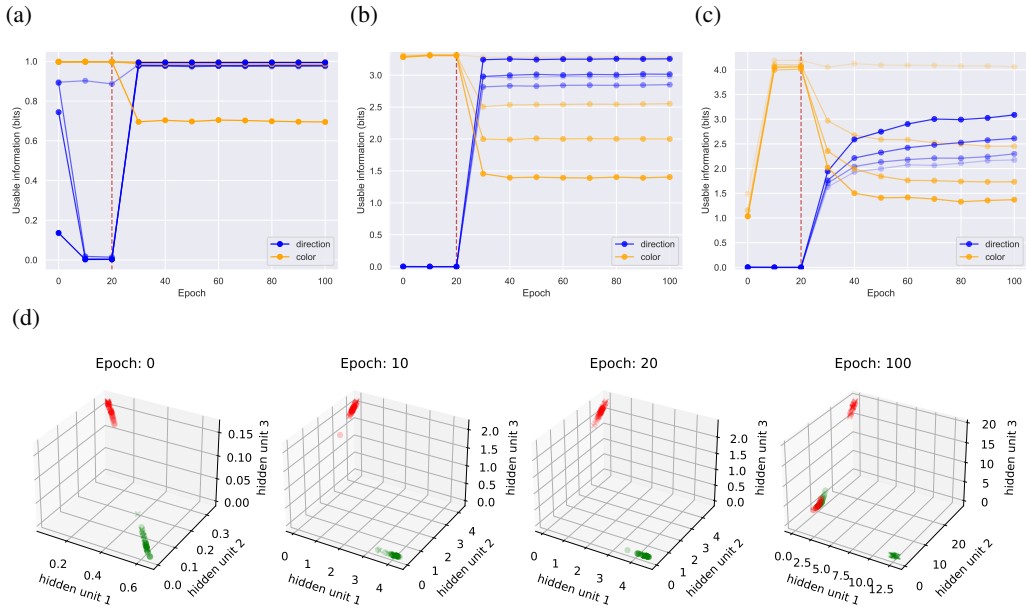

Figure 3: Usable color and direction information in a network through training following pretraining the network to output color, not direction. Pretraining occurred for the first 20 epochs, indicated by the dashed red line. Subsequently, the network was trained to output direction, as in Fig 2. **(a)** Usable information for Small FC trained on the $N = 2$ CB task. Usable color information increased in training, and decreased when the loss function changed. However, the asymptotic representation is not minimal. **(b)** Medium FC trained on $N = 10$ CB task. Similarly, the network formed a representation of color during pretraining, but the asymptotic representation is not minimal. **(c)** Medium FC trained on $N = 20$ checkerboard task. **(d)** Visualization of the Small FC network in (a) showing that an optimal representation is not formed. The asymptotic representation in the last area has separate representations for red and green crosses. These should be overlapping in a minimal representation.

## 3.2 SGD with non-random initializations may not form minimal representations in the CB task

Implicit regularization in SGD is hypothesized to result in minimal representations through compression of irrelevant input information, also called a "forgetting" phase (Shwartz-Ziv & Tishby, 2017; Achille & Soatto, 2018; Achille et al., 2019). We tested this hypothesis by initializing networks with significant color information, and subsequently performing SGD on the CB task. We then evaluated whether SGD resulted in networks with minimal color representations. We initialized networks by pretraining the network to output the color decision for 20 epochs, which required the network to represent color information. After 20 epochs, we reverted to training the CB task, where only the direction decision was reported. Since the learning rate was kept constant, the pretrained weights can be viewed as a different initialization in parameter space for the modified task.

We found that the resulting representations were not minimal for the $n = 2$ checkerboard case (Fig 3a). This result also held for the CB task with $n = 10$ and $n = 20$ (Fig 4b,c). While we observed some compression of usable color information through training, asymptotic representations had significantly greater than zero color information. In Fig 4b, we observed all layers had more usable color information than the direction information in the first layer. The network therefore solved the task using an alternative representation that was not minimal. We visualized the activations corresponding to the asymptotic non-minimal representations of Small FC in Fig 3d. In the early epochs the red and green points converge (both crosses and dots) as a result of successful pretraining. However, when we trained the CB task starting at epoch 20, the representations changed. While the dot clusters for red and green checkerboards are overlapping, the cross clusters are not. This representation is not minimal as color information can be decoded above chance.

These results show that the initialization affects the asymptotic representation of neural networks. SGD, under particular initializations, may not lead to minimal representations of task inputs. This suggests there is a trade-off between minimal representations and using existing representations present in the initial weights. Initial structure in the network representations from pretraining, such as the separation of the red and green crosses in the last layer representation, was maintained even when performing SGD to train a different task. Together, these results suggest that while SGD compresses representations towards minimality, it finds a solution that is functionally related to the initial representation. This may correspond to a optima in the neighborhood of the initialization. Further, we found pretraining on the color choice led to worse generalization performance (Fig 4).

## 4   Discussion

We introduced a notion of the usable information in the representation, which reflects the amount of information that can be extracted by a learned decoder. This definition is appealing, in part, due to its flexibility. For instance, if it is important to understand how accessible the information is to a linear decoder, it suffices to apply our formulation of usable information with a linear decoder trained with cross-entropy loss. In contrast, if the goal is extract all the information present in a representation, regardless of how accessible this information is, one can train a high capacity nonlinear decoder. Since neural networks are powerful function approximators, as the function approximation improves, the decoder will approach the optimal decoder. In this case, the usable information approaches Shannon mutual information, as the lower bound becomes tight (Section B.1). Future theoretical and empirical work should investigate the tightness of this bound and its dependence on training parameters.

In our case, we used a relatively small nonlinear neural network as the decoder, which provided insight into the evolution of optimal representations through training on simple tasks inspired by neuroscience literature. These tasks allowed us to show that random initializations and the implicit regularization from SGD play an important role in learning minimal sufficient representations. Notably, we found that a non-random initialization, corresponding to pretraining on a related but different task, led to solutions that were less likely to form optimal representations and had worse generalization performance. Future work should evaluate the learning dynamics of more realistic network architectures on more challenging tasks.

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

# A  Additional experiments

## A.1  Relationship between pretraining, minimality, and generalization in the CB task

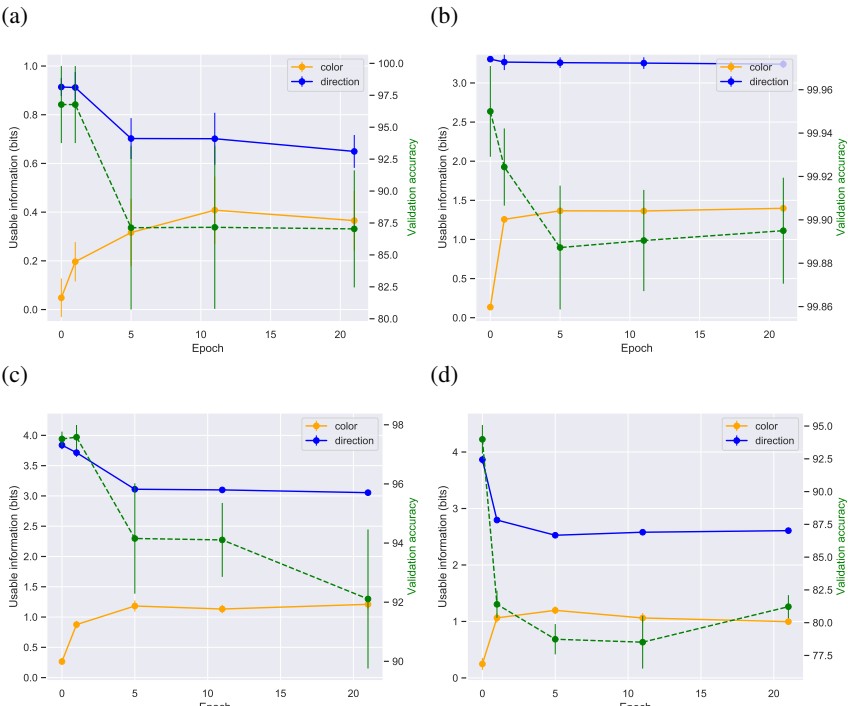

Figure 4: **(a)** Final usable information and accuracy as a function of pretraining epoch for the CB task ($n = 2$) averaged over $8$ random initializations. **(b)** Final usable information and accuracy as a function of pretraining epoch for the CB task ($n = 10$) averaged over $8$ random initializations. **(c)** Final usable information and accuracy as a function of pretraining epoch for the CB task ($n = 20$) averaged over $8$ random initializations. **(d)** Final usable information and accuracy as a function of pretraining epoch for the CB task ($n = 25$) averaged over $8$ random initializations. Error bars show the S.E.M.

Our results show that the minimality of network representations, and therefore solutions, depend on initialization. All trained networks (for $n$ larger than 2), however, achieved zero training error. A natural question to ask is how do the resulting representations affect generalization performance?

To answer this, we varied the number of epochs that we pretrained the CB tasks of $n = 2$, $n = 10$, and $n = 20$ classes, and quantified the usable color and direction information, as well as the trained network's test accuracy to understand how the network generalizes (Fig. 4). We found a positive correlation with the minimality of the representation and generalization performance: networks with less usable color information achieved higher test set accuracy. This was true regardless of the number of classes, but the effect was more pronounced (in terms of absolute difference in accuracy) when the network did not solve the task perfectly without pretraining. We note that regardless of how long the networks were pretrained for, the networks were subsequently trained for the same number of epochs (80), with the same learning rate throughout training. One interpretation is that when using existing structure to solve the task, the network learned a suboptimal solution to solving the task, increasing the chance of overfitting.

## A.2  Binary MNIST classification

Our earlier analyses used relatively simple tasks where it is straightforward to characterize relevant and irrelevant representations. But do our findings that the resulting representations found by SGD changed based upon the initialization extend to a more realistic and complex task? To this end, we trained a network to predict whether digits from MNIST were either even or odd. One solution

(a)               (b)

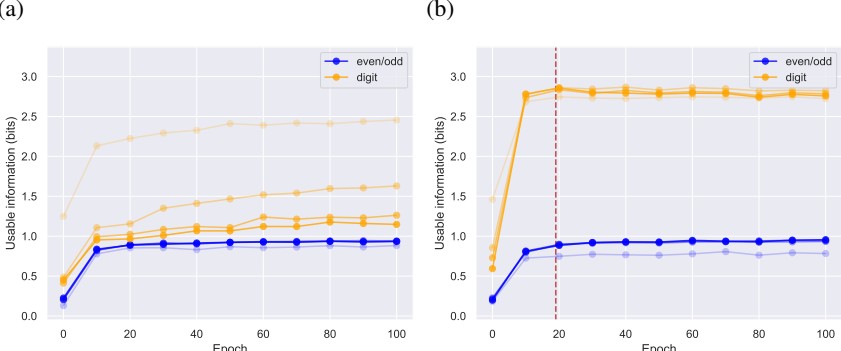

Figure 5: MNIST even/odd classification. **(a)** A Large FC architecture was trained to predict whether MNIST digits were even or odd. The resulting representation contained a nearly minimal representation (with 1 bit of usable digit information, corresponding to whether the digit was even or odd). **(b)** The network was pretrained for the first 20 epochs to output the correct digit and subsequently trained to predict whether the digits were even or odd. SGD did not result in minimal network representations, with representations containing almost 3 bits of usable digit information. We also did not observe noticeable compression of digit information.

the network could find is to group features corresponding to even and odd digits, without explicitly representing the digits. This minimal solution would have 1 bit of digit information (corresponding to whether a digit is even or odd, but no other information). An alternative solution is to represent each digit and learn a classifier that can group the digits into even or odd. This representation is not minimal, and would have closer to the maximal 3.32 bits of usable digit information.

When we trained a Large FC architecture to predict whether digits were even or odd, we found that the resulting representation was nearly minimal (Fig 5a). We are not claiming that the output layer has no additional input related information, but rather that the digit cannot be decoded from the representation.

We then changed the task, so that the network was first asked to output the correct digit and then, the task was switched so that the network was only asked to output whether the digit was even or odd. We pretrained the network for 20 epochs to output the correct digit, resulting in nearly 3 bits of usable digit information (Fig 5b). After pretraining, we subsequently trained the network to only perform the even/odd classification task. We found the asymptotic representation had little to no compression of digit information. Instead, it solved the task, with approximately the same amount of digit information as in the initialization. This suggests that SGD reused features present in the representation and arrived at an alternative task solution.

## B Proofs

### B.1 Usable information lower bounds the mutual information

The entropy of a distribution is defined as

$$H(x) = \mathbb{E}_{x \sim p(x)} \left[ \log \frac{1}{p(x)} \right]. \tag{2}$$

The mutual information, $I(X;Y)$, can be written in terms on an entropy term and as conditional entropy term:

$$I(Z;Y) = H(Y) - H(Y|Z). \tag{3}$$

We want to show that:

$$I(Z;Y) \geq I_u(Z;Y) := H(Y) - L_{CE}(p(y|z), q(y|z)) \tag{4}$$

It suffices to show that:

$$H(Y|Z) \leq L_{CE} \tag{5}$$

where $L_{CE}$ is the cross-entropy loss on the test set. For our study, $H(Y)$ represented the known distribution of output classes, which in our case were equiprobable.

$$H(Y|Z) := \mathbb{E}_{(z,y) \sim p(z,y)} \left[ \log \frac{1}{p(y|z)} \right] \tag{6}$$

$$= \underbrace{\mathbb{E}_{(z,y) \sim p(z,y)} \left[ \log \frac{1}{q(y|z)} \right]}_{\text{cross-entropy loss}} - \underbrace{\mathbb{E}_{z \sim p(z)} \left[ \text{KL}(p(y|z)||q(y|z)) \right]}_{\geq 0}, \tag{7}$$

$$\leq \mathbb{E}_{(z,y) \sim p(z,y)} \left[ \log \frac{1}{q(y|z)} \right] := L_{CE} \tag{8}$$

To approximate $H(Y|Z)$, we first trained a neural network with cross-entropy loss to predict the output, $Y$, given the hidden activations, $Z$, learning a distribution $q(y|z)$. The KL denotes the Kullback-Liebler divergence. We multiplied (and divided) by an arbitrary variational distribution, $q(y|z)$, in the logarithm of equation 6, leading to equation 7. The first term in equation 7 is the cross-entropy loss commonly used for training neural networks. The second term is a KL divergence, and is therefore non-negative. In our approximator, the distribution, $q(y|x)$, is parametrized by a neural network. When the distribution $q(y|z) = p(y|z)$, our variational approximation of $H(Y|Z)$, and hence approximation of $I(Z;Y)$ is exact (Barber & Agakov, 2003; Poole et al., 2019). Recently, such variational approximations to mutual information have been viewed as a meaningful characterization of representations in deep networks, and the theoretical underpinnings of this approach is beginning to be investigated (Xu et al., 2020; Dubois et al., 2020).

## C    Experimental details

### C.1    Details of neural network for usable information

To estimate usable information, we computed the cross-entropy loss of a decoder q(y|z) that predicts $Y$ from $Z$. The decoder was a three layer neural network, with 128, 64, and 32 units per layer, with leakyRelu activations (slope = 0.2), batchnorm and dropout ($p = 0.7$). At each epoch, 1250 training samples were generated and supplied to the decoder, along with either the corresponding correct direction or color choice. We evaluated the cross-entropy loss on 3750 test samples to minimize overfitting. We trained the network for 100 epochs using a learning rate of 0.5 for 'Medium FC' and 0.05 for 'Small FC.' For the 'Large FC' used in MNIST experiments, we used a learning rate of 0.005 for 1000 epochs.

### C.2    Checkerboard Task description

Following the conventions of Kleinman et al. (2019), we modeled the CB task (Fig 1a), inputting the checkerboard color and target configuration to a neural network that outputted the direction choice (Fig 1b). We minimized the cross-entropy loss of the network output and the ground truth output. We extended the checkerboard task to the *n* checkerboard task by increasing the number of checkerboards. Each target was 1 out of the $n$ colors, with the targets forming an 'n-polygon'. The correct direction corresponded to the direction of the target corresponding to the color of the checkerboard. We specified the color of each target as a one hot encoding, and the color of the checkerboard as a one hot encoding. Noise with mean 0 and standard deviation of 0.1 was added to the checkerboard inputs. The targets and checkerboard color inputs were concatenated to form an input vector. The correct direction of the target was the output.

### C.3    Details of experiments

The following are the hyperparameters used in our experiments. We trained three different network architectures, 'Small FC': 5 layers, with $10 - 7 - 5 - 4 - 3$ units in each layer, 'Medium FC': $100 - 20 - 20 - 20$, and 'Large FC': $1000 - 20 - 20 - 20$. We trained networks using SGD with a constant learning rate throughout training.

**FC Small,** $n = 2$:

- batch size: 32, learning rate: 0.05, number of data samples: 10000 (90% train, 10% validation)

**Medium FC, $n = 10$:**

- batch size: 64, learning rate: 0.5, number of data samples: 25000 (90% train, 10% validation)

**Medium FC, $n = 20$:**

- batch size: 128, learning rate: 0.5, number of data samples: 50000 (90% train, 10% validation)

**Medium FC, $n = 25$:**

- batch size: 128, learning rate: 0.5, number of data samples: 75000 (90% train, 10% validation)

**Large FC, (MNIST):**

- batch size: 128, learning rate: 0.05

### C.4 Definition of relevant and irrelevant information

In the CB task, the color of the checkerboard and target configuration (inputs) are necessary to determine the correct direction to reach (output). While both a color and direction decision are made, after the direction is determined, the color decision no longer needs to be represented: the network can generate the correct output with only the direction representation. Formally, the output $y$ is conditionally independent of the color representation, $Z_c$, given the direction representation $Z_d$ (i.e., $y \perp\!\!\!\perp (Z_c, Z_t)|Z_d$, as illustrated by the graph in Fig 1b). Hence, given a representation of the direction choice, the color choice (and target configuration) no longer needs to be represented. We emphasize that, in general, the output is not independent of the color representation and target configuration representation $Z_t$, i.e., $y \not\perp\!\!\!\perp (Z_c, Z_t)$, hence information about the dominant color of the checkerboard is necessary to compute $y$. When this conditional independence holds, we call the conditionally independent variable "irrelevant." We therefore refer to color choice as "irrelevant" and the direction choice as "relevant." We study how these components evolve together throughout training.

### C.5 Binary MNIST task description

We trained the FC Mnist architecture to output whether the digit was even or odd. Accordingly, a minimal representation should only encode whether the digit was even or odd, and not the particular digit.

## D  Additional Figures

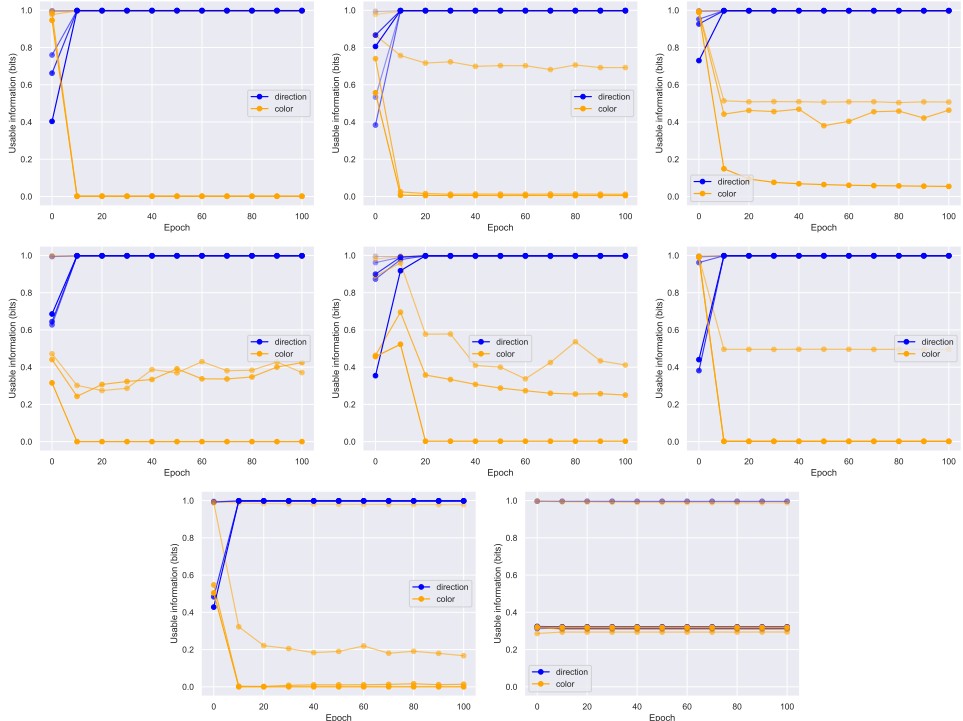

Figure 6: Evolution of usable information for eight random initializations for the $n = 2$ CB task.

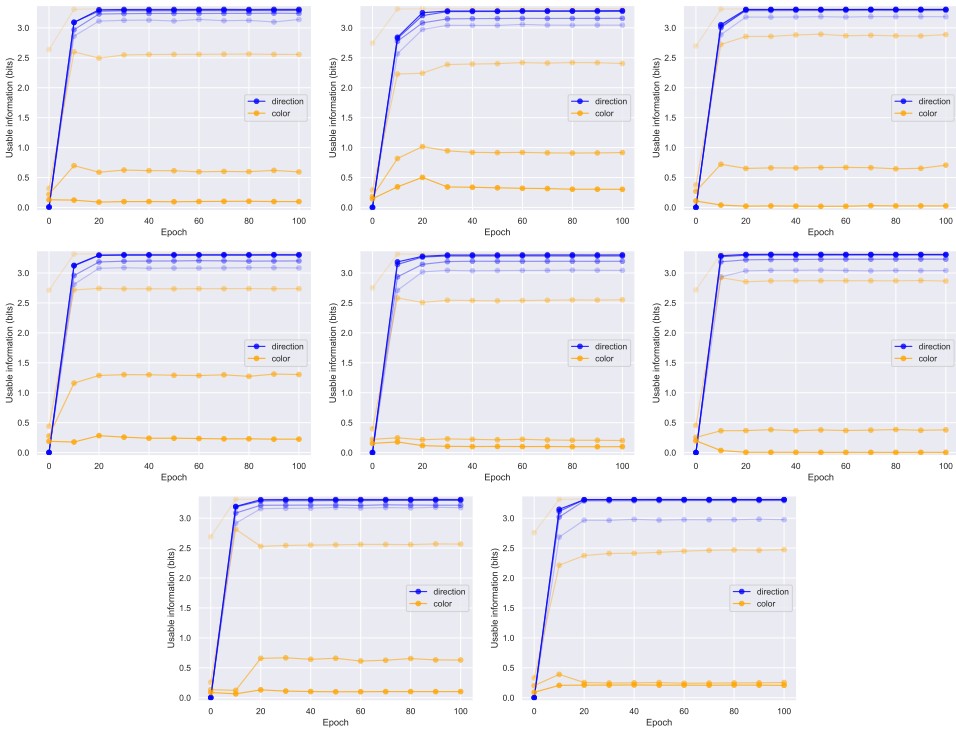

Figure 7: Evolution of usable information for eight random initializations for the $n = 10$ CB task.

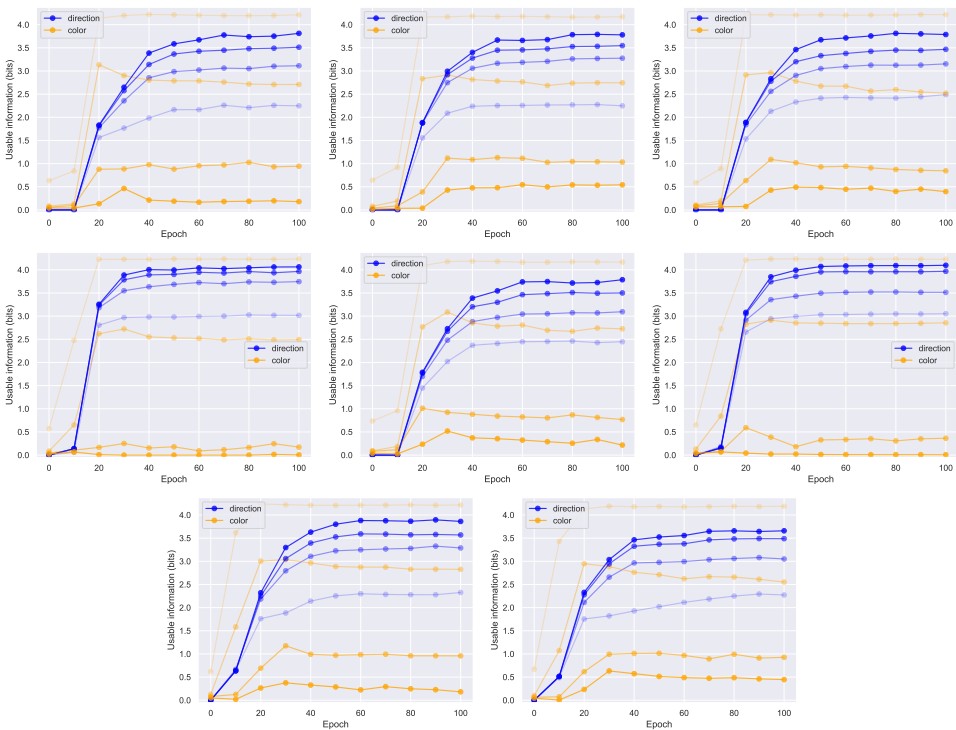

Figure 8: Evolution of usable information for eight random initializations for the $n = 20$ CB task.

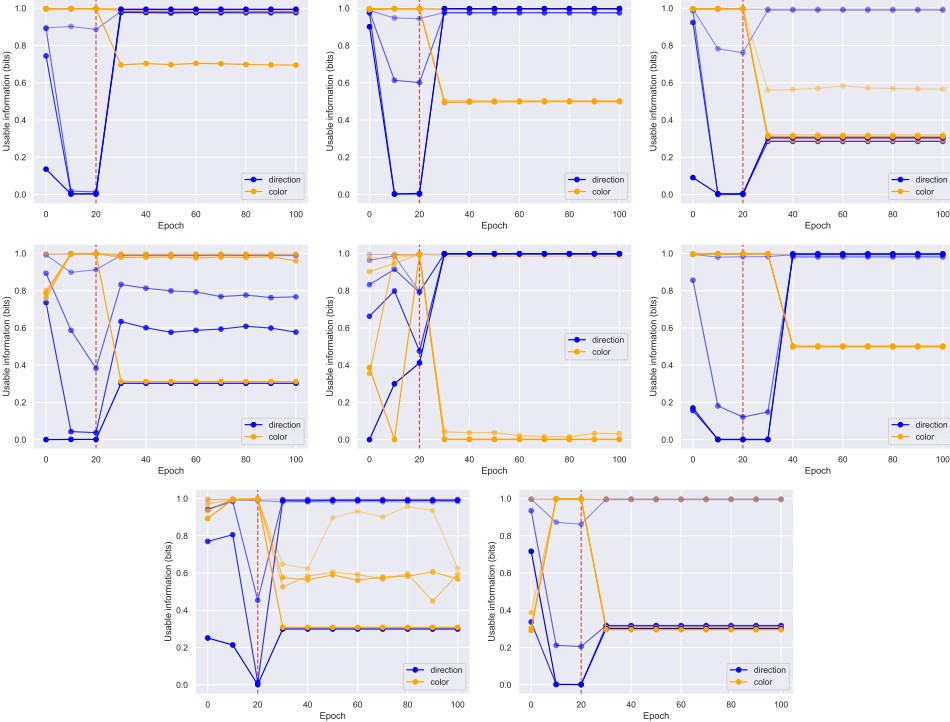

Figure 9: Evolution of usable information for eight random initializations for the $n = 2$ CB task with 20 epochs of pretraining. If the the usable information was negative, indicating that the decoder overfit, we set the usable information to 0. Note that this occurred for a very small number of points.

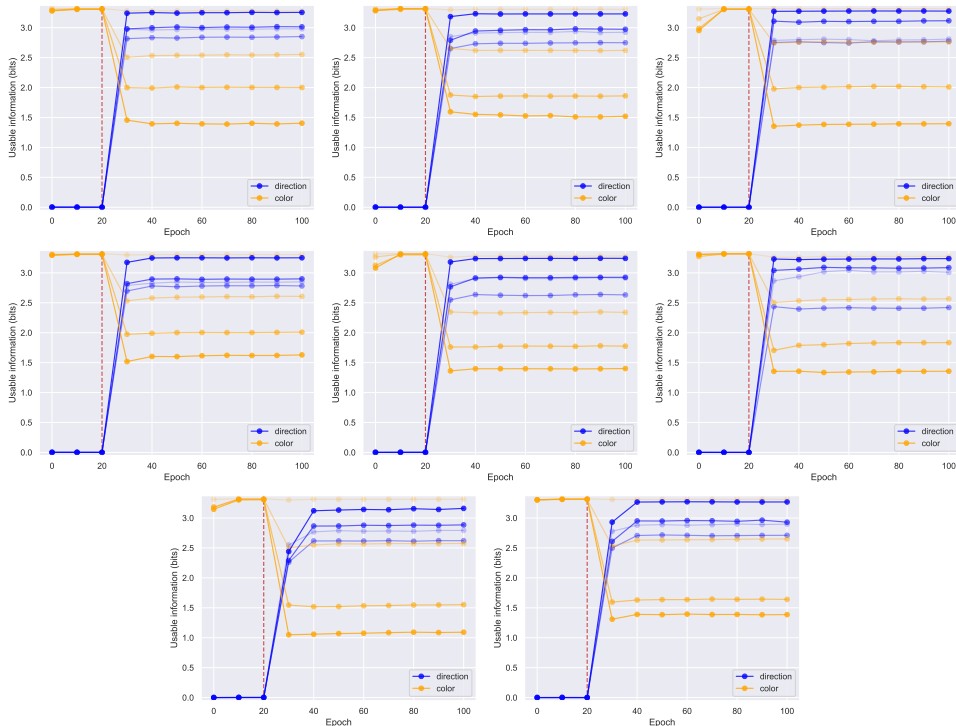

Figure 10: Evolution of usable information for eight random initializations for the $n = 10$ CB task with 20 epochs of pretraining.

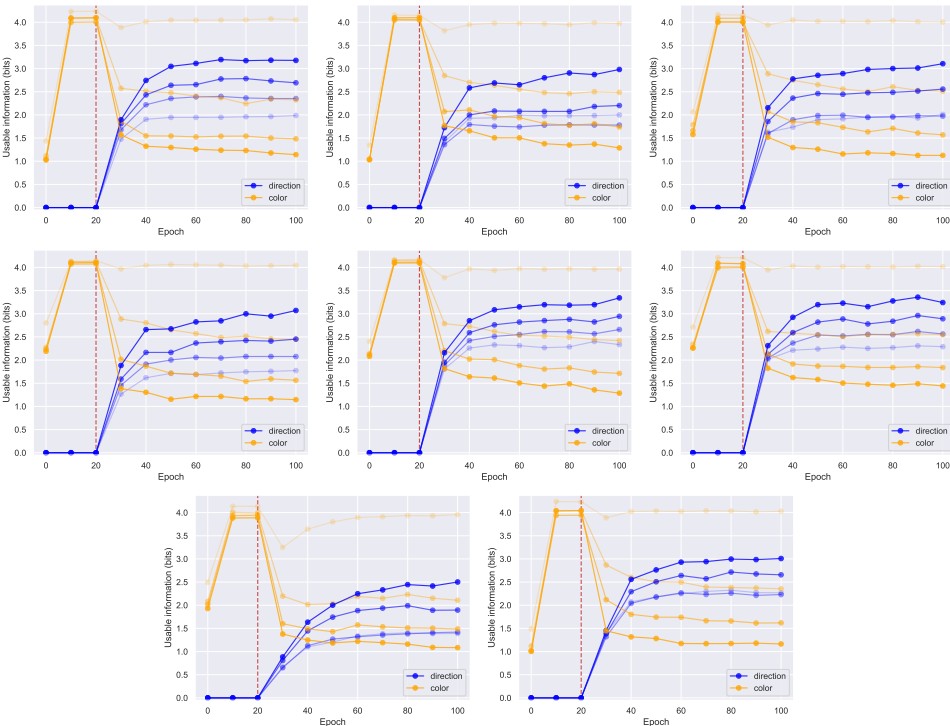

Figure 11: Evolution of usable information for eight random initializations for the $n = 20$ CB task with 20 epochs of pretraining.

