# OpenReview forum: "Usable Information and Evolution of Optimal Representations During Training"
_NeurIPS.cc/2020/Workshop/SVRHM — SVRHM@NeurIPS Poster_

### Official Review · AnonReviewer3 · 2020-10-28
**Are Stochastic Gradient Descent and random initialization necessary for learning minimal representations?**

**Rating:** 5
**Confidence:** 5

**Review:**

In this paper, the authors investigate the hypothesis that Stochastic Gradient Descent (SGD) with random initialization leads to learning minimal representations that are required for the training task. To show that, different fully connected neural networks with three different sizes were trained on a simple color discrimination task. The trained network should learn to report the direction that corresponds to the right choice of color for every trial. Due to the random color/direction assignment in every data sample (every trial of the experiment), the authors claimed that a minimal representation at the output layer shouldn’t carry any information about the color. As the results showed, throughout the training, the color information decreases for deeper layers, which is suggested as an evidence of learning minimal representations. The authors asked very interesting and important questions in the paper, and took a reasonable approach to answer the questions, but there are a few issues that need to be addressed to clarify the reasoning behind their claims.


Major comments:

I have three main concerns about the interpretation of the results of this paper:

First - the task that is used in this paper is not the right task for the purpose of the study. In the CB task, even though we do not expect color information represented at the output layer,  it does not mean that color information is irrelevant for the task. Hence, the notion of information ‘compression’ might not hold here. In order to choose the right direction, the network should properly combine color information and output configuration to generate the correct output. Therefore, the color information is not discarded throughout the network, but rather, projected to the correct output direction. In an analogy with the brain, we could say that the early layers of the trained ANNs have learned the sensory processing (discriminating color) while the deeper layers are responsible for sensorimotor transformation (output direction). The differential information content of the deep and superficial layers of the trained networks in figure 3 and 4 also support this interpretation. Thus, the alternative interpretation, instead of minimal representation learning, is that the networks learn the correct sensorimotor transformation. This is consistent with older studies that showed that ANNs are capable of learning sensorimotor transformations similar to those observed in parietal and association brain areas (e.g. Zipser and Anderson, Nature, 1988). Another variation of the CB task, which could better serve the purpose of this study, is the one used by Mante and Sussillo et al (2013). In this version of the task, unlike the CB task in the paper, red/green dots also move in a certain direction. In every trial, subjects would be asked either about the dots’ color or motion direction. When asked about the dots’ motion direction, the color information would be irrelevant and needed to be discarded during processing. I believe the notion of minimal representation would make more sense in the context of this version of the task.

Second - the paper’s main claim is that SGD has a critical role in learning minimal representations. This hypothesis implies that if we use another learning rule, the minimal representations (as claimed in the paper) will not be learned, or will not be learned as efficiently. This implication needs to be tested explicitly. Namely, the networks should be trained with an alternative learning rule (e.g. GD or Hebbain learning, etc.) and show that the same pattern of usable information is not achieved with alternative learning rules.

Third - I am not sure about the interpretations suggested by the pretraining experiments. We already knew that once a pretraining task interferes with a new task, as is the case in this paper, the pretrained weights need to be at least partially forgotten by the network before performing well in the new task. Pretraining is usually used when the representations required for the second task are partly consistent with the representations learned by the pretraining task. Even though the color representation learned by the pretraining task is useful for the second task, the output projections to different output directions need to be learned from scratch (and possibly from a worse initialization). Given the simplicity of the CB task, random initialization leads to faster learning (random initialization is probably closer to an optimum loci), and at the end, the pretraining doesn’t help much. However, this does not mean that random initialization is needed for minimal representation learning, but rather shows that the chosen pretraining task is not useful for the CB task.

Minor comment:
Reference to figure 2 in the text appears later than figures 3 and 4. Why is it placed before these two figures in the paper?

---

### Official Review · AnonReviewer2 · 2020-10-29
**Learning minimal representations and initialization**

**Rating:** 6
**Confidence:** 4

**Review:**

This paper attempts to provide an information-theoretic measure that would evaluate how optimal a representation learned by a deep network is. The authors start by noting that mutual information can be degenerate, and propose usable information as measured by a second decoder network.
I think this is an interesting idea, though one could argue that it is a bit weird to define usable information in one network using a second network – seems like a circular argument.
Additionally, the authors do show that usable information defined in this way in the limit becomes mutual information, so it is again difficult to not worry about the degeneracy.

It is important to note that the figures after references (integral to the main part of the paper) take the paper to 8 pages + references + appendix, above the recommended limit of the workshop.

Here are some additional remarks and questions for the authors:
- It could be interesting to see how conv layers would impact the results here.
- In Figure 3, the Medium FC Net has some usable color information – is this because the network used to measure this information simply lacks the capacity to analyze the bigger networks?
- I find the minimal representation experiment with pretraining interesting, in that it shows that non-random initializations lead to keeping representations that are irrelevant to the new task. Is this however not particularly surprising, unless catastrophic forgetting was to be involved?
- In section 3.3 it is not entirely clear what kind of pretraining took place. Is it the same as in the previous section?

Despite these concerns and questions, I think this is an interesting direction to consider.

---

### Public Comment · ~Michael_Kleinman2 · 2020-12-08
**Revision uploaded**

We thank the reviewers for their thoughtful comments. We have incorporated their feedback into an updated manuscript, now posted. In particular, we have condensed the material so that the figures are included in the main text, as well as expanded on the properties of usable information that make it appealing for investigating learning dynamics. We also agree with the reviewers that it is important to evaluate the learning dynamics on different architectures (convolutional networks) and more challenging tasks. We have commented on this in the discussion, and is something we are working on.

---

### Decision · Program_Chairs · 2020-11-02

Accept (Poster)